# Biological Effects of EGCG@MOF Zn(BTC)_4_ System Improves Wound Healing in Diabetes

**DOI:** 10.3390/molecules27175427

**Published:** 2022-08-25

**Authors:** Song Li, Jing Yan, Qiangqiang Zhu, Xinxiang Liu, Senlin Li, Shenhou Wang, Xuanjun Wang, Jun Sheng

**Affiliations:** 1College of Science, Yunnan Agricultural University, Kunming 650201, China; 2Key Laborotory of Puer Tea Sciencs, Ministry of Education(YNAU), Yunnan Agricultural University, Kunming 650201, China; 3Agro-Products Processing Research Institute, Yunnan Academy of Agricultural Sciences, Kunming 650201, China

**Keywords:** EGCG, MOF, wound healing, diabetes

## Abstract

Tea contains high levels of the compound epigallocatechin gallate (EGCG). It is considered an important functional component in tea and has anti-cancer, antioxidant, and anti-inflammatory effects. The eight phenolic hydroxyl groups in EGCG’s chemical structure are the basis for EGCG’s multiple biological effects. At the same time, it also leads to poor chemical stability, rendering EGCG prone to oxidation and isomerization reactions that change its original structure and biological activity. Learning how to maintain the activity of EGCG has become an important goal in understanding the biological activity of EGCG and the research and development of tea-related products. Metal–organic frameworks (MOFs) are porous materials with a three-dimensional network structure that are composed of inorganic metals or metal clusters together with organic complexes. MOFs exploit the porous nature of the material itself. When a drug is an appropriate size, it can be wrapped into the pores by physical or chemical methods; this allows the drug to be released slowly, and MOFs can also reduce drug toxicity. In this study, we used MOF Zn(BTC)_4_ materials to load EGCG and investigated the sustained release effect of EGCG@MOF Zn(BTC)_4_ and the biological effects on wound healing in a diabetic mouse model.

## 1. Introduction

Tea is one of the most popular drinks in the world—far exceeding beer, coffee, and carbonated drinks— and is well known in the forms of green, black, or oolong tea. Among these forms, green tea has had the most obvious impact on human health [1,2,3,4]. The production of green tea is to cut, twist, and dry freshly picked young tea tree leaves to produce a dry and stable product. The enzymes responsible for decomposing the pigments in the young leaves are destroyed in the process of curing, so the leaves remain green during the subsequent rolling and drying process. This production method preserves the natural polyphenol content in tea, which are the compounds responsible for its biological activity. Most green tea polyphenols (GTP) are flavonols, usually called catechins. There are four main types of catechins in green tea: epicatechin (EC), epicatechin gallate (ECG), epigallocatechin(EGC), and epigallocatechingallate (EGCG). The most abundant is EGCG [5,6,7].

The ester compound synthesized by the combination of gallocatechol and gallic acid is EGCG, which has a molecular weight of 453 kDa. EGCG is a four-ring structure containing eight phenolic hydroxyl groups. These structural characteristics imbue strong antioxidant properties. Studies have shown that EGCG can directly inhibit the expression of inflammatory factors such as tumor necrosis factor (TNF)-α, interleukin (IL)-6, and IL-lβ, thereby achieving anti-inflammatory effects. EGCG undergoes auto-oxidation to generate hydrogen peroxide that has long-term antioxidant activity [8,9]. EGCG can combat the harmful effects of many potentially pathogenic bacteria [10,11,12]. For example, EGCG is very effective in inhibiting the growth of Staphylococcus aureus and methicillin-resistant Staphylococcus aureus (MRSA) [13,14,15]. Among catechins, EGCG reacts effectively with most ROS (reactive oxygen species). However, due to its unique chemical structure, EGCG itself is also very unstable and is easily affected by factors such as temperature, metal ions, enzymes, and pH. It is prone to oxidation and isomerization reactions, thereby changing its original structure and biological activity.

At present, protein nanoparticles, polysaccharide nanoparticles, liposomes, etc., are used as delivery bodies to encapsulate drugs, which can prevent the isomerization and oxidation reaction of EGCG, improve its slow release and stability, and realize targeted delivery, thus expanding the application range of EGCG. However, EGCG has poor lipid solubility, resulting in poor permeability in cells and low absorption rate, resulting in a low oral utilization rate of liposomes in clinical treatment. In addition, liposomes are easily damaged by a weak acid environment and digestive enzymes in the process of digestion, resulting in the leakage of the load. Therefore, liposomes as delivery carriers of EGCG still need further research. The stability of EGCG-CS nanoparticles in the intestine was affected by the porosity of polysaccharides, low embedding rate, and core material [16].

In recent years, porous materials have attracted much attention in the fields of materials, chemistry, physics, and others because of their simple preparation process, large specific surface area, easy-to-optimize structure, and orderly porous structure. Metal–organic frameworks (MOFs) are a new type of porous material that combines inorganic and organic components through physical or chemical action. MOF materials not only have a crystal structure similar to regular pores of zeolite molecular sieve but also have a higher specific surface area than traditional porous materials such as polymer, an inorganic polymer, and carbon-based materials. In addition, it contains organic components, which makes its structure tailorable, designable, adjustable pore size, and functional pore surface. MOF materials have a rich structure, and the hole size and type can be designed according to project needs. MOFs could be used to develop slow-release drugs and even address environmental pollution and energy issues [17,18].

MOF materials are prepared by mixing metal ions with organic complexes in a solvent, heating the mixture to generate and grow crystal nuclei, and combining the metal ions with small molecular substances in the solvent to form a competitive relationship with the organic complexes. After washing and drying, the small molecules of the solvent can be separated from the metal sites, thereby forming unsaturated metal sites, which enables adsorption or catalysis. The MOF material itself has a porous structure. When the size of the pores is larger than a drug, that drug can be loaded into the pore by physical or chemical means. Once in a specific environment, the drug can be released slowly over time, which can reduce drug toxicity and side effects.

MOF Zn(BTC)_4_ is a new MOF material synthesized by our research group. It was chosen as a model MOF because its synthesis was simple and scalable, and the toxicity of the ligand H3BTC was reported to be low [19].

The most serious complications of diabetes occur after long-term treatment. In clinical research, the development of chronic inflammation is an important sign of spontaneous diabetes. The level of inflammation is correlated with diabetes complications [20]. Slow wound healing is one of the common characteristics of subjects with diabetes. Wounds in these patients are difficult to heal due to chronic inflammation and high blood sugar [21]. The presence of chronic inflammation in diabetic wounds is manifested by an increase in pro-inflammatory factors [22]. Compared with normal wounds, diabetic wounds exhibit obvious high expression of inflammatory factors, including IL-1β, TNF-α, and others [23].

EGCG itself has a strong anti-inflammatory effect and can promote wound healing in diabetic mice [24]. However, because of the unsaturated bond in the structure of EGCG, its chemical stability is poor, and the instability of EGCG also leads to a decrease in its bioavailability, which limits its application in the food and pharmaceutical industries. Therefore, in this study, the streptozotocin (STZ)-induced diabetic mouse model was used as the main research object to explore the effects of EGCG and EGCG@MOF Zn(BTC)_4_ on wound healing in diabetic mice. We explored whether EGCG@MOF Zn(BTC)_4_ can normally release EGCG, reduce its degradation rate, and prolong its biological effects [25].

## 2. Result and Discussion

### 2.1. General Information

The following materials were obtained from the indicated vendors: tribenzoic acid (Shanghai Darui Fine Chemicals Co., Ltd. Shanghai, China), zinc nitrate (Aladdin), absolute ethanol (Tianjin Fengchuan Chemical Reagent Technology Co., Ltd. Tianjin, China), N,N-dimethylamide (Sinopharm Chemical Reagent Co., Ltd. Company, Shanghai, China), methanol (Tianjin Fengchuan Chemical Reagent Technology Co., Ltd.), EGCG (purity above 98%) (Chengdu Purifa Technology Development Co., Ltd. Chengdu, China), RAW264.7 cells (Kunming Institute of Animal Cell Bank), lipopolysaccharide (Sigma), TNF-α kit (Beijing Sizhengbai Biological Co., Ltd. Beijing, China), STZ (Sigma, St Louis, MO, USA), ICR mice (Changzhou Cavens, Changzhou, China), sodium carboxymethyl cellulose (Aladdin), Ma Song (Masson) Staining kit (Beijing Zhongshan Jinqiao Biotechnology Co., Ltd. Beijing, China), IL-1β kit (Beijing Sizhengbai Biotechnology Co., Ltd. Beijing, China), and IL-6 kit (Beijing Sizhengbai Biotechnology Co., Ltd. Beijing, China). The data for crystals were collected on a Bruker APEX-II CCD diffractometer (MoKa, l ¼ 0.71073 Å) at 298(2) K. Infrared spectra (KBr pellets) were taken on a Bruke Tensor 27 FTIR spectrometer in the range of 4000–400 cm^−1^. Morphology was recorded on a FlexSEM 1000 scanning electron microscope. Microscopic characterization was finished by Tianqing microscope (optical magnification: 50×–1000×, accuracy: 0.001 mm).

### 2.2. Synthesis of MOF Zn(BTC)_4_

Dissolve Zn(NO_3_)_2_•6H_2_O in 60 mL of ultrapure water to yield a concentration of 6.7 mmol, stir it evenly, add methanol and trimesic acid on this basis, and finally heat it in an oven at 100 °C for 24 h. After the heating process is over, the oven is kept closed until the temperature is naturally cooled down to room temperature. The cooled sample was filtered; cleaned; and dried under vacuum at 60 °C to obtain white, rectangular, powdery crystals. Finally, the sample was collected, sealed, and stored for later use.

### 2.3. Preparation of EGCG@MOF

Weigh 20 mg of MOF Zn(BTC)_4_ into a 2 mL Eppendorf (EP) tube, add 1 mL of 140 mg/mL EGCG solution, mix well, place in a shaker at 4 °C, and rotate slowly for 12 days. Afterward, take out the EP tube and centrifuge at 4 °C, 4500 r/min for 10 min, discard the supernatant as much as possible, add 2 mL of absolute ethanol to resuspend EGCG@MOF, repeat the centrifugation twice, discard both supernatants, and wash it as much as possible to remove the remaining EGCG on the surface of the MOF. Add 1 mL of anhydrous ethanol–water solution to the EP tube, place it in a freezer at −80 °C until frozen, then place it in a freeze dryer to produce dry EGCG@MOF powder.

### 2.4. Detection of Drug Loading of EGCG@MOF

Accurately weigh the mass of EGCG@MOF in the EP tube, add 1 mL of acidified phosphate-buffered saline (PBS) solution to resuspend it, and place it in a 37 °C water bath for ultrasound for 1 h to completely release the EGCG in the MOF. Filter it with a 0.45 μm pore size filter, use high-performance liquid chromatography (HPLC) to detect the concentration of EGCG, and calculate the drug loading amount of EGCG for the MOF.

### 2.5. HPLC Detection of EGCG Concentration

A sample filtered through a 0.45 μm pore was used to determine the EGCG content in the sample using HPLC. The platform was equipped with G1311B Quat pump (1 mL per minute), G1329B autosampler, G1316A TCC column oven (40 °C), and G1314F ultraviolet detector (280 nm) (Agilent 1260 series HPLC system, Agilent, Santa Clara, CA, USA). The sample volume was 10 μL, and the HPLC experiment was controlled with 1260LC Agilent ChemStation software (Agilent Technologies). A C18 ODS column (ZORBAX SB-C18 4.6 mm 250 mm, 5 μm, Agilent) was used to complete the separation. The mobile phases used in the HPLC analysis were solvents A (100% acetonitrile) and B (0.03% trifluoroacetic acid), which were filtered through a 0.45 μm filter. Within 25 min, mobile phase A increased from 10% to 60%.

### 2.6. Cell Processing and Grouping

RAW264.7 cells were digested, counted with a cell counter, and inoculated with 10 mL of culture medium at the number of five million cells per plate. Then the plates were put in a 37 °C incubator for 24 h. Afterward, the culture medium was discarded, the cells were washed once with PBS, and serum-free high-glycemic culture medium was added for starvation conditions and cultured overnight.

After overnight starvation, the cells were given fresh serum-free medium, and all EGCG@MOF Zn(BTC)_4_ groups were added according to the release concentration of EGCG to a final concentration of 10 μg/mL and placed in a 37 °C incubator for 30 min. The dosage of the MOF Zn(BTC)_4_ group was added according to EGCG@MOF Zn(BTC)_4_ group. After 30 min, lipopolysaccharide (LPS) solution with a concentration of 200 EU/mL was added to the culture medium, and the plates were placed in a carbon dioxide incubator at 37 °C for another 3 h and 30 min before collecting the cell supernatant and extracting the protein. There were five groups of cells used in the experiments: Control, LPS, MOF Zn(BTC)_4_, EGCG@MOF Zn(BTC)_4_, and LPS+EGCG@MOF Zn(BTC)_4_.

### 2.7. Establishment of Diabetic Mouse Model

All ICR mice fasted for 15 h with free access to water. The mice were then weighed using an electronic scale, the average weight was calculated, and the cages were randomly divided according to the rules of consistent weight average. According to our preliminary studies and reference to other experimental methods, a 1% STZ solution was prepared for intraperitoneal injection of ICR mice at a dose of 100 mg/kg. The injected mice were placed back into the cage, allowed to eat and drink freely, and raised normally. Their blood glucose levels were tested once a week for 14 days. Reaching a blood glucose level >20 mmol/L and then maintaining it for more than 2 weeks was considered a successful model for diabetic mice.

### 2.8. Establishment of a Wound Healing Model

The successful ICR diabetic mouse model and normal ICR mice were used as the research objects for the skin wound model. After using an electric razor to completely shave the dorsal fur, a depilatory cream was applied to completely remove the remaining hair. After rinsing off the depilatory cream with warm water, the skin was dried with absorbent paper, and the mice were placed back in the cage under a warm lamp. Because the depilatory cream can irritate mouse skin of the mice, the wounds were made the next day.

The mice were anesthetized by inhaling ether from their mouths. The dorsal area was washed with 75% ethanol, and an 8 mm biopsy device was used to remove a piece of skin on the left and right sides of the mouse’s back. A steel ruler was used as an indicator to mark the areas, treat the wounds with the compounds, and determine how photos were taken for the experimental records.

### 2.9. Wound Healing Experiment Grouping and Treatment

There were five experimental groups: normal, diabetes, MOF Zn(BTC)_4_, EGCG@MOF Zn(BTC)_4_, and EGCG. The normal group consisted of control ICR mice, and the other four groups were randomly divided diabetic model mice (7 mice/group). All mice were reared under the same environmental conditions with ad libitum access to food and water. All wound model mouse dressings are prepared with 1% carboxymethyl cellulose (CMC) solution. Diabetic and normal mice were treated with 1% CMC solution as a placebo. Mice in the EGCG@MOF Zn(BTC)_4_ group were treated with EGCG@MOF Zn(BTC)_4_ dressings with an EGCG concentration of 20 μg/mL. The amount of MOF Zn(BTC)_4_ groups added is equivalent to EGCG@MOF Zn(BTC)_4_ group addition amount. The mice in the EGCG group were treated with 20 μg/mL EGCG.

The drugs in all treatment groups were given once every 4 days. The experimental period was 15 days. All treatments were given at 10 a.m., and the wounds were photographed and measured every day. The images were analyzed ImageJ software (National Institutes of Health) to measure wounds, and these data were used to calculate the healing rate and determine if there were significant differences between the different treatment groups.

The wound healing rate was calculated as: 1–wound area on day n(1,2,3…)/initial wound area.

### 2.10. Wound Sampling Treatment

Depending on the results of the wound healing experiment in mice, different time points were selected for tissue extraction with an 8 mm biopsy sampler. After tissue collection, the samples were washed in cold PBS to remove blood, dried with filter paper to remove surface water, and then half were placed in formalin for at least 24 h. The other half were placed in liquid nitrogen for quick freezing, and then transferred to an −80 °C freezer for storage until subsequent experiments. Mouse skin tissue preserved in liquid nitrogen is used for enzyme-linked immunosorbent assay to detect related inflammatory factors.

## 3. Results and discussion

### 3.1. Structural Characterization of MOF Zn(BTC)_4_

The structure of MOF Zn(BTC)_4_ was characterized by crystal diffraction. The crystallographic data of MOF Zn(BTC)_4_ are listed in Table 1, and the selected bond lengths and angles are listed in Table 2. Figure 1 respectively shows the SBU image and 3D frame structure of this crystal.

CCDC: 1539458 for compound 1.

### 3.2. Load Capacity of MOF Zn(BTC)_4_ to Hold EGCG

Due to MOF’s dual characteristics of high drug loading and controllable drug release, it has quickly become a research hotspot. Studies have shown that a small amount of mesoporous MOFs can carry a large drug load, and MOFs with larger pore diameters have more application value [26]. Multi-level MOFs with micropores, mesopores, and macropores can simultaneously carry out drug molecules of different sizes [27].

We first determined the peak of EGCG as a reference standard with HPLC, and the peak time of EGCG was 9.910 min (Figure 2A). After soaking MOF Zn(BTC)_4_ in 50 mg/mL EGCG solution, the sample was rotated on a shaker at 4 °C for 8 days, then centrifuged twice at 4 °C with absolute ethanol to clean the surface of MOF from residual EGCG, and then ethanol was used to resuspend in an aqueous solution prior to freeze-drying to produce MOF material loaded with EGCG. The loaded EGCG@MOF was added to PBS and ultrasonicated at 37 °C for 1 h to completely release the EGCG in the MOF material. After filtration, the EGCG content was detected by HPLC. As shown in Figure 2B, the EGCG peak existed in the liquid phase spectrum of MOF Zn(BTC)_4_ at 9.856 min, indicating that MOF Zn(BTC)_4_ can load EGCG. Under these conditions, EGCG@MOF Zn(BTC)_4_ loading was 56.247 mg/g.

Next, we optimized EGCG loading into MOF Zn(BTC)_4_ to EGCG. To increase drug loading, we performed a time course experiment. We weighed several pieces of 20 mg of MOF Zn(BTC)_4_ into a 2 mL EP tube, added 1 mL of 50 mg/mL EGCG, mixed it upside down, placed it in a 4 °C shaker, and rotated it slowly. Every two days, one piece was removed, centrifuged, washed to remove surface EGCG, then frozen and dried to obtain MOF Zn(BTC)_4_-EGCG. Next, 1 mL PBS was added before ultrasonication for 1 h. The resulting solution was filtered prior to HPLC detection of EGCG to calculate the amount of EGCG that can be loaded by MOF Zn(BTC)_4_. The amount of EGCG loaded in EGCG@MOF Zn(BTC)_4_ increased with time and reached the maximum on the 12th day (Figure 3). The highest loading amount of EGCG in EGCG@MOF Zn(BTC)_4_ reached 94.134 mg/g on day 12 and then decreased, which means that EGCG@MOF Zn(BTC)_4_ loaded with EGCG gradually accumulated with the increase in time, when the cumulative EGCG reached the maximum, the loaded EGCG will fall off and then eventually re-accumulate. Therefore, the amount of EGCG in MOF Zn(BTC)_4_-EGCG will fluctuate within a certain range.

To further increase the loading capacity of EGCG@MOF Zn(BTC)_4_, we prepared different concentrations of EGCG (20, 30, 40,…, 200 mg/mL) for EGCG@MOF Zn(BTC)_4_ loading. Briefly, we weighed 20 mg of EGCG@MOF Zn(BTC)_4_ into a 2 mL EP tube, added 1 mL of EGCG of different concentrations, and placed the tubes on a shaker at 4 °C with slow rotation for 12 days. They were then centrifuged to wash away residual EGCG on the surface of EGCG@MOF Zn(BTC)_4_, followed by freezing and freeze-drying to obtain EGCG@MOF Zn(BTC)_4_. As described above, 1 mL PBS was added prior to 1 h ultrasonication. After filtering the solution, HPLC was performed to detect the concentration of EGCG and the amount of EGCG loaded in MOF Zn(BTC)_4_. The amount of EGCG loaded in EGCG@MOF Zn(BTC)_4_ varied with EGCG concentration (Figure 3B). When the EGCG concentration was 140 mg/mL, EGCG@MOF Zn(BTC)_4_ reached a maximum of 236.227 mg/g.

### 3.3. Chemical Characterization of MOF Zn(BTC)_4_ and EGCG@MOF Zn(BTC)_4_

We observed MOF Zn(BTC)_4_ and EGCG@MOF Zn(BTC)_4_ powder grossly and microscopically and analyzed the morphological changes of the MOF Zn(BTC)_4_ powder after loading EGCG.

In Figure 4A, the left and right sides show MOF Zn(BTC)_4_ and EGCG@MOF Zn(BTC)_4_ powder, respectively. MOF Zn(BTC)_4_ is a white powdery solid, while MOF Zn(BTC)_4_ is soaked in an EGCG solution and then freeze-dried. The color becomes slightly yellow after this process. MOF Zn(BTC)_4_ itself is relatively finely broken and loose, and the powder form is an acicular solid when observed by the naked eye. When piled together, it appears relatively dense, while MOF Zn(BTC)_4_ is soaked in EGCG solution. The color turns to pale yellow, the powder state has been destroyed, and the needle-like shape observed with the naked eye becomes more finely broken. When piled together, it appears fluffy; when packed in an EP tube, it is prone to hang on the walls of the tube. Figure 4B shows microscope images of the MOF Zn(BTC)_4_ and EGCG@MOF Zn(BTC)_4_ powders on the left and right, respectively. The MOF Zn(BTC)_4_ powder is clearly visible and relatively dispersed with a fusiform structure. After filling the MOF Zn(BTC)_4_ powder with EGCG and then freeze-drying, the original fusiform structure of MOF Zn(BTC)_4_, becomes more finely fragmented and is easier to pile together. The solid materials are crushed and easily accumulate with obvious adhesion.

The infrared (IR) spectrum analysis comparison of EGCG@MOF Zn(BTC)_4_ and MOF Zn(BTC)_4_ is shown in Figure 5A. The vibration peaks at 3387 cm^−1^, 1619 cm^−1^, 1557 cm^−1^, 1443 cm^−1^, 1372 cm^−1^, 761 cm^−1^, and 721 cm^−1^ did not change. However, compared with the absorption peak at 1110 cm^−1^ in the MOF Zn(BTC)_4_ spectrum, new absorption peaks appeared at 1127 cm^−1^, 1145cm^−1^, and 1059 cm^−1^ in the EGCG@MOF Zn(BTC)_4_ spectrum. This may be the stretching vibration generated by the C-O single bond in EGCG. These new absorption peaks likely appeared because introducing EGCG into MOF Zn(BTC)_4_ changed the IR spectrum.

MOF Zn(BTC)_4_ and EGCG@MOF Zn(BTC)_4_ morphologies were assessed by scanning electron microscopy (SEM) at different magnifications (Figure 5B). MOF Zn(BTC)_4_ crystals had a massive appearance, while the EGCG@MOF Zn(BTC)_4_ crystal was more finely fragmented. After filling with EGCG, the structure, size, and crystal arrangement of MOF Zn(BTC)_4_ were altered to certain degrees. As shown in the X-ray diffraction (XRD), results of MOF Zn(BTC)_4_ and EGCG@MOF Zn(BTC)_4_ were also compared (Figure 5C). When MOF Zn(BTC)_4_ was filled with EGCG, the intensities of some characteristic peaks of MOF Zn(BTC)_4_ changed significantly, but the peak shapes of MOF Zn(BTC)_4_ were basically the same. After loading, the skeleton structure of MOF Zn(BTC)_4_ was basically maintained.

PBS was used to prepare 0.5 mg/mL EGCG that was distributed into several 2 mL EP tubes, then 1 mL EGCG solution was added to each EP tube, and PBS was used to prepare 0.5 mg/mL EGCG@MOF Zn(BTC)_4_ according to the EGCG concentration. Next, 1 mL of EGCG@MOF Zn(BTC)_4_ solution was dispensed into several 2 mL EP tubes, the tubes were placed in a 37 °C drying oven, and 1 EP tube per group was removed every hour for 12 h. After that, one EP tube was taken out every 2 h for each group. Following filtration, the concentration of EGCG was detected by HPLC (Figure 6). The concentration of EGCG gradually decreased over time. At 24 h, EGCG has basically been completely degraded at a very fast rate. Conversely, EGCG@MOF Zn(BTC)_4_ can gradually release EGCG and maintain the concentration within a certain range. The concentration of EGCG released by EGCG@MOF Zn(BTC)_4_ also increased as a whole. It gradually decreased over time but to a much smaller degree than EGCG. At 120 h, the concentration of EGCG released by EGCG@MOF Zn(BTC)_4_ was still maintained at 0.27 mg/mL.

At 10 h, the concentration of EGCG released by EGCG@MOF Zn(BTC)_4_ was 0.38 mg/mL, and the concentration of EGCG alone was 0.27 mg/mL. At 20 h, the corresponding values were 0.49 and 0.13 mg/mL. At 26, 28, and 30 h, the contents of EGCG fluctuated because of the difference in the content of EGCG@MOF Zn(BTC)_4_ in each EP tube. It can be clearly seen from the figure that the concentration of EGCG decreased linearly with time in both groups, but MOF Zn(BTC)_4_ can stabilize the concentration of EGCG to a certain extent and slow its degradation. This may be due to the existence of pores in MOF Zn(BTC)_4_ itself, which protects EGCG to a certain extent. Based on these results, we can conclude that EGCG@MOF Zn(BTC)_4_ can slow down the degradation rate of EGCG, thereby prolonging its action time.

### 3.4. Research on Anti-Inflammatory Effects of EGCG@MOF Zn(BTC)_4_ Targeting the Notch Signaling Pathway

The inflammatory response produced by macrophages can be attenuated by regulating Notch signaling, which further regulates the state of macrophages. Researchers have shown that blocking Notch1 protein production can reduce the activation level of macrophages [28]. We performed western blotting (WB) to detect the effect of EGCG on Notch protein secretion after 30 min and 3 h to assess whether EGCG can inhibit macrophage secretion of Notch-related proteins.

After LPS stimulation, the expression level of Notch1 protein expressed in macrophages did not increase significantly (Figure 7A). However, the amounts of Notch1 protein expressed in the EGCG@MOF Zn(BTC)_4_+LPS and EGCG@MOF Zn(BTC)_4_ groups were significantly lower than the control group after 3 h. The results indicate that EGCG@MOF Zn(BTC)_4_ can inhibit the expression of Notch1 protein over time, but there was no obvious change after 30 min.

Notch2 protein expression in macrophages was significantly increased after LPS stimulation at 30 min and 3 h stimulation. After 30 min incubation with EGCG@MOF Zn(BTC)_4_, Notch2 protein levels were not dramatically changed, but they were significantly reduced after 3 h stimulation. Macrophages treated with EGCG@MOF Zn(BTC)_4_+LPS had higher Notch2 protein levels than those treated with EGCG@MOF Zn(BTC)4, indicating that EGCG@MOF Zn(BTC)_4_ can dampen inflammation in macrophages over time.

To further illustrate the effect of EGCG@MOF Zn(BTC)_4_ on Notch signaling, we once again used WB to detect the effect of EGCG@MOF Zn(BTC)_4_ on Notch signaling pathway-related proteins at 30 min and 3 h. After 30 min of LPS treatment, Cleaved-Notch1 levels increased; it can be clearly seen that after treatment with EGCG@MOF Zn(BTC)_4_, the expression levels of Cleaved-Notch1 and Hes1 proteins in macrophages were significantly reduced (Figure 7B). EGCG@MOF Zn(BTC)_4_ had a significant inhibitory effect on Cleaved-Notch1 and Hes1 protein levels. When comparing the 30 min and 3 h timepoints, Hes1 protein levels increased with time. Moreover, after EGCG@MOF Zn(BTC)_4_ treatment, Hes1 protein expression after 3 h was significantly lower than in the 30 min treatment group. This also shows that EGCG@MOF Zn(BTC)_4_ can reduce the inflammation state of macrophages. Again, this was time-dependent.

Collectively, these results demonstrate that EGCG@MOF Zn(BTC)_4_ can reduce the inflammatory response of macrophages caused by the Notch signaling pathway by reducing Notch protein secretion. This shows that EGCG@MOF Zn(BTC)_4_ can facilitate the anti-inflammatory effect of EGCG.

After 3 h LPS (200 EU/mL) stimulation of mouse RAW264.7 macrophages, the cell supernatants of each group were collected, and the inflammatory factor TNF-α was detected. The results showed that TNF-α expression increased compared with the control group (Figure 4). After a 30 min treatment with EGCG@MOF Zn(BTC)_4_, LPS-induced TNF-α secretion by macrophages was significantly reduced. This shows that EGCG@MOF Zn(BTC)_4_ can effectively reduce the expression of inflammatory factors in mouse macrophages.

### 3.5. EGCG@MOF Zn(BTC)_4_ Improved the Stability of EGCG Action

Establishing an animal model of diabetes is important for investigating diabetes and related complications. Common diabetes models include high-fat diet-induced, obesity-induced (ob/ob food-induced), genotype (db/db), and pharmacologically induced. Diabetes models successfully induced by drugs have the advantages of stability and fast modeling, so they are widely used in numerous studies of diabetes and its complications [29]. The ICR mouse is commonly used as an animal research object in the study of diabetes and its complications [30]. Here we injected ICR mice with STZ and performed continuous blood glucose testing to confirm the successful establishment of diabetic mice that were used in wound model experiments.

Studies have shown that EGCG can reduce inflammation in cells and tissues, suggesting that EGCG exerts anti-inflammatory effects [31]. In order to clarify whether EGCG@MOF Zn(BTC)_4_ can improve EGCG stability and allow it to reduce inflammation, wounds were made on the dorsal skin of diabetic mice, and EGCG@MOF Zn(BTC)_4_ or EGCG were applied once every four days. Images and measurements were taken daily to observe the effects of EGCG@MOF Zn(BTC)_4_ and EGCG on wound healing in diabetic mice.

Wound healing evolution is shown in Figure 8A. To better explore the effects of EGCG@MOF Zn(BTC)_4_, we generated a line chart of the wound non-healing rate.

Compared with the model group, normal mice showed significant differences on days three to nine (Figure 8B). At these time points, the proportions of unhealed wound area in normal mice were 50.5%, 43.41%, 35.96%, 31.05%, 26.56%, 23.01%, and 17.88%. The wound non-healing rates of diabetic mice were significantly higher than that of normal mice at 85.35%, 73.59%, 60.87%, 52.52%, 40%, 29.86%, and 24.7%. This shows that inducing experimental diabetes can indeed slow down the rate of wound healing.

EGCG@MOF Zn(BTC)_4_ and EGCG were applied on the first day and every four days thereafter. Figure 8B shows that treatment groups showed significant differences on days four to nine. EGCG@MOF Zn(BTC)_4_ more obviously improved wound healing in diabetic mice. This suggests that EGCG@MOF Zn(BTC)_4_ can improve the stability of EGCG. At these time points, the non-healing rates of the EGCG@MOF Zn(BTC)_4_ treatment group were 51.63%, 42.12%, 32.52%, 25.04%, 22.82%, and 18.06%, which were close to the healing rate of the normal group. The non-healing rates of the wounds in the EGCG-treated group were 79.27%, 67.62%, 53.27%, 46.67%, 39.92%, and 34.93%, which were similar to those of mice in the model group.

Compared with the Normal group, the MOF Zn(BTC)_4_ group showed significant differences on days three to eight (Figure 8C). At these time points, the unhealed wound areas in MOF Zn(BTC)_4_-treated mice were 85.38%, 72.08%, 68.37%, 56.94%, and 42.22%, respectively. The diabetic mouse wound model did not differ significantly from the model group after applying MOF Zn(BTC)_4_, but the results were significantly different from those in the Normal group. This shows that MOF Zn(BTC)_4_ alone does not significantly promote wound healing in mice. Compared with the EGCG@MOF Zn(BTC)_4_ group, mice treated with MOF Zn(BTC)_4_ also showed a significant difference on days three to eight. This indicates that it is the EGCG residing in EGCG@MOF Zn(BTC)_4_ that promotes wound healing. The slow-release effect promotes anti-inflammatory effects and wound healing in diabetic mice. These results suggest that EGCG@MOF Zn(BTC)_4_ improves the stability of EGCG.

### 3.6. EGCG@MOF Zn(BTC)_4_ Promotes Skin Wound Healing in Diabetic Mice

Wound healing is a complex process that is influenced by many cytokines. Inflammation is the main factor that slows diabetic wound healing [29], which is also hindered by abnormal cellular responses. To investigate the effect of cytokines on wound healing in diabetic mice, we measured some of the cytokines released from the wound sites during the healing process. Researchers have shown that secretion of the inflammatory factor IL-1β is clearly related to slow wound healing. Reducing the release of IL-1β protein in tissue can accelerate the healing speed of diabetic wounds [31]. The pro-inflammatory factor IL-6 is also an important factor in this process.

Enzyme-linked immunosorbent assays were used to detect IL-1β and IL-6 in the wound skin tissue of diabetic mice on day nine (Figure 9).

Compared with the normal group, levels of IL-1β released in the wounds of the diabetic model mice were increased, indicating a greater degree of inflammation. IL-1β levels in the MOF Zn(BTC)_4_ and model groups were similar, suggesting that MOF Zn(BTC)_4_ itself cannot produce an anti-inflammatory effect. After EGCG_@_MOF Zn(BTC)_4_ treatment, the release of IL-1βwas significantly reduced in diabetic mice; it was almost as low as that measured in the normal group, and it was significantly lower than that of mice in the EGCG group. Again, this indicates that MOF Zn(BTC)_4_ improves the stability of EGCG. Compared with normal skin tissue, the release level of the pro-inflammatory factor IL-6 was significantly increased in diabetic wound tissue, and the results were similar to those of IL-1β. Overall, wounds of diabetic mice treated with EGCG@MOF Zn(BTC)_4_ showed obviously lower expression of inflammatory factors.

To further observe the effect of EGCG@MOF Zn(BTC)_4_ on skin wound healing in diabetic mice, we performed hematoxylin and eosin (H&E) staining to detect re-epithelialization. To investigate if slow wound healing in diabetic mice was improved by EGCG@MOF Zn(BTC)_4_ treatment on a microscopic level, skin wound tissues were collected on day nine for paraffin sectioning and conventional H&E. The differences in tissue re-epithelization are shown in Figure 10. The results indicate that EGCG@MOF Zn(BTC)_4_ promotes wound healing in diabetic mice.

The non-epithelialized area of skin tissue in the normal group was significantly reduced compared with the model group, and it was clearly observed whether there was a completely healed area in the model group of mice. The MOF Zn(BTC)_4_ group did not have significant differences compared with the model group, and EGCG-treated mice did not show a significant improvement in re-epithelialization. The EGCG@MOF Zn(BTC)_4_ treatment group and EGCG treatment. The mice in the group are close to the re-epithelialization of the skin tissue of the model group. This shows that the administration of EGCG alone every four days does not improve wound healing in diabetic mouse models.

Compared with the model and EGCG treatment groups, re-epithelialization was significantly improved in the EGCG_@_MOF Zn(BTC)_4_ treatment group and was close to that of the normal group. Compared with the EGCG alone, EGCG@MOF Zn(BTC)_4_ treatment significantly improves the re-epithelialization of skin wound tissue in diabetic mice. This again supports the hypothesis that EGCG@MOF Zn(BTC)_4_ can improve EGCG stability.

Extracellular matrix (ECM) is a macromolecular substance synthesized by cells in the dermis. It is distributed between cells or on their surfaces, where it plays an indispensable role in wound healing. ECM components mainly include proteoglycans, proteins, and polysaccharides, and collagen is an important component and marker of ECM. The substances in ECM form a complex network structure that supports and connects the tissue structure, regulates tissue development, and impacts the physiological activities of cells. ECM formation is an important physiological reaction in the process of wound healing, and a higher level of ECM formation can promote skin wound healing. As chronic diabetic wounds heal, different types of infiltrating cells appear at the skin wound site, which reduces ECM formation and slows wound healing [32].

Researchers have shown that decreased collagen deposition will slow down the rate of wound healing [33]. To observe new ECM formation and assess the effect of EGCG_@_MOF Zn(BTC)_4_ treatment, we used Masson’s trichrome staining to observe collagen deposition levels on day nine of wound healing (Figure 11).

Masson staining showed that collagen deposition (blue area) in the wounds of the diabetes model group was significantly lower than that of the normal group. Compared with the model group, the MOF Zn(BTC)_4_ group did not show significant differences, and the mice in the EGCG treatment group did not show a significant improvement in collagen deposition. The mice in the MOF Zn(BTC)_4_ and EGCG groups had collagen deposition similar to the model group. This suggests that EGCG administration every four days does not improve wound healing in a diabetic mouse model.

Compared with the model and EGCG treatments, collagen deposition was significantly improved in the EGCG@MOF Zn(BTC)_4_ treatment group, with similar re-epithelialization as observed in the normal group. These results indicate that EGCG@MOF Zn(BTC)_4_ can increase collagen deposition in skin wounds of diabetic mice compared with EGCG treatment alone, thus significantly improving ECM formation. This also indicates that EGCG@MOF Zn(BTC)_4_ can improve the stability of EGCG for treatment purposes.

## 4. Conclusions

Our results demonstrate that MOF Zn(BTC)_4_ can be loaded with EGCG. After optimizing the loading conditions, each gram of MOF Zn(BTC)_4_ can be loaded with 236.227 mg EGCG. Chemical characterization of MOF Zn(BTC)_4_ using IR spectroscopy showed that a new absorption peak appeared after adding EGCG. XRD analysis confirmed that the skeleton structure of MOF Zn(BTC)_4_ filled with EGCG is basically maintained, and EGCG@MOF Zn(BTC)_4_ can slowly release EGCG, which reduces the degradation rate of EGCG. By slowly releasing EGCG, EGCG@MOF Zn(BTC)_4_ can reduce the expression of related inflammatory factors and significantly inhibit the inflammatory response in LPS-induced macrophages. In addition, EGCG@MOF Zn(BTC)_4_ can significantly inhibit the expression of proteins related to the Notch signaling pathway and reduce the expression of inflammatory TNF-α. Wound healing experiments in diabetic mice revealed that adding EGCG to MOF Zn(BTC)_4_ can slow down the degradation rate of EGCG, enabling slower release and prolonging the time for EGCG to function. EGCG@MOF Zn(BTC)_4_ treatment can significantly reduce inflammation in the skin wound tissue of diabetic mice. Compared with the skin wound tissue of control diabetic mice, the re-epithelialization and collagen accumulation levels of the EGCG group were significantly increased. Collectively, the results demonstrate that EGCG_@_MOF Zn(BTC)_4_ has the ability to release EGCG slowly, which slows its degradation and prolongs its action time.

Although MOFs have many significant advantages, there are still many challenges in clinical application. Compared with other nanocarriers, the development of MOFs as drug carriers is still in its infancy. Many problems need further optimization and research, such as the toxicity of metal ions, the degradability of MOFs, and the improvement of the biocompatibility of MOFs with the body. However, we have reason to believe that MOFs will have broad development prospects in the future.

## Figures and Tables

**Figure 1 molecules-27-05427-f001:**
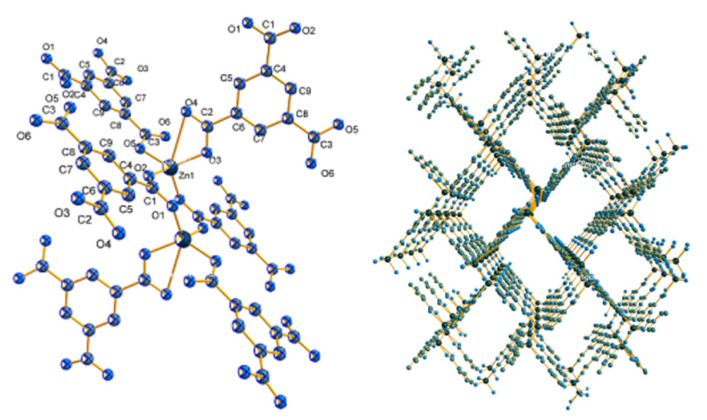
SBU image and 3D framework of MOF Zn(BTC)_4_.

**Figure 2 molecules-27-05427-f002:**
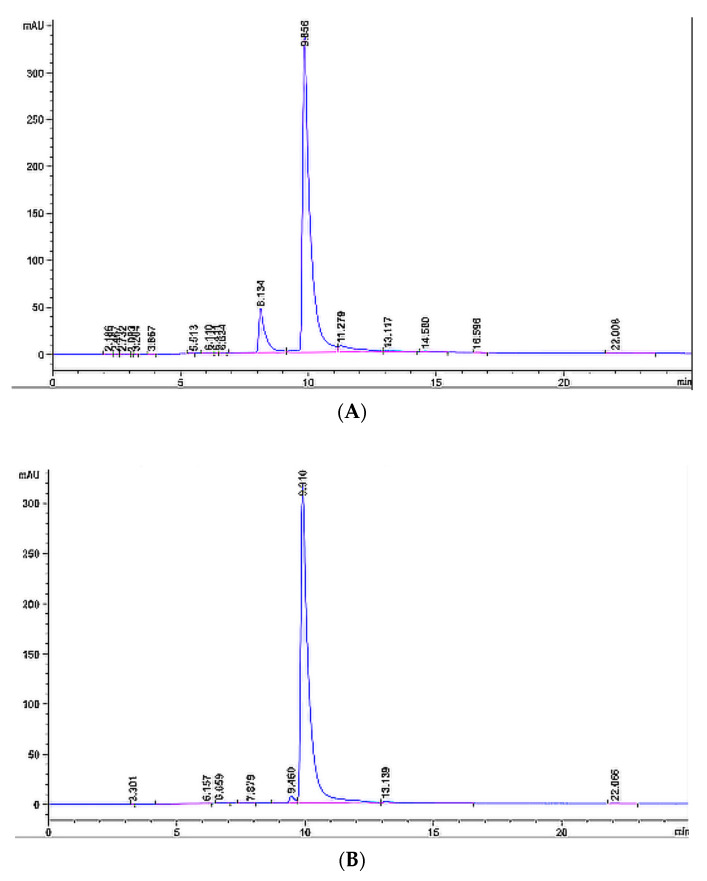
HPLC(High Performance Liquid Chromatography) detection of EGCG. (**A**) HPLC spectrum of EGCG standard; (**B**) MOF Zn(BTC)_4_ loading of EGCG.

**Figure 3 molecules-27-05427-f003:**
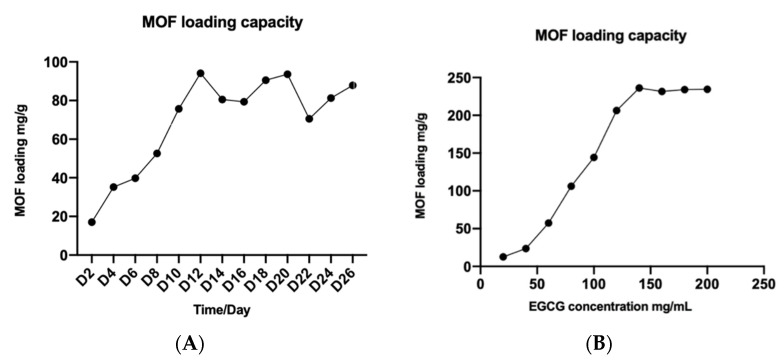
Optimizing MOF Zn(BTC)_4_-EGCG loading. (**A**) The loading capacity of MOF Zn(BTC)_4_ on EGCG over time; (**B**) The influence of different EGCG concentrations on EGCG@MOF Zn(BTC)_4_ loading capacity.

**Figure 4 molecules-27-05427-f004:**
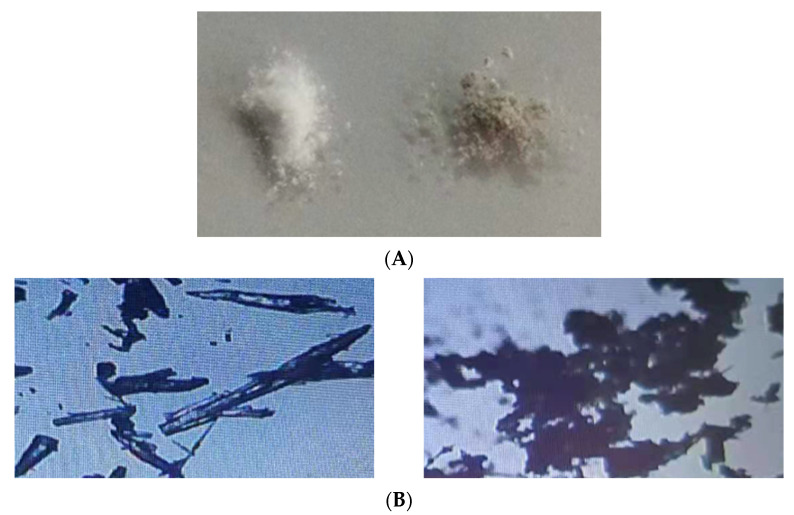
Characterization of MOF Zn(BTC)_4_ and EGCG@MOF Zn(BTC)_4_. (**A**) Gross images of MOF Zn(BTC)_4_ and EGCG@MOF Zn(BTC)_4_; (**B**) Microscope images of MOF Zn(BTC)_4_ and EGCG@MOF Zn(BTC)_4_ (40×).

**Figure 5 molecules-27-05427-f005:**
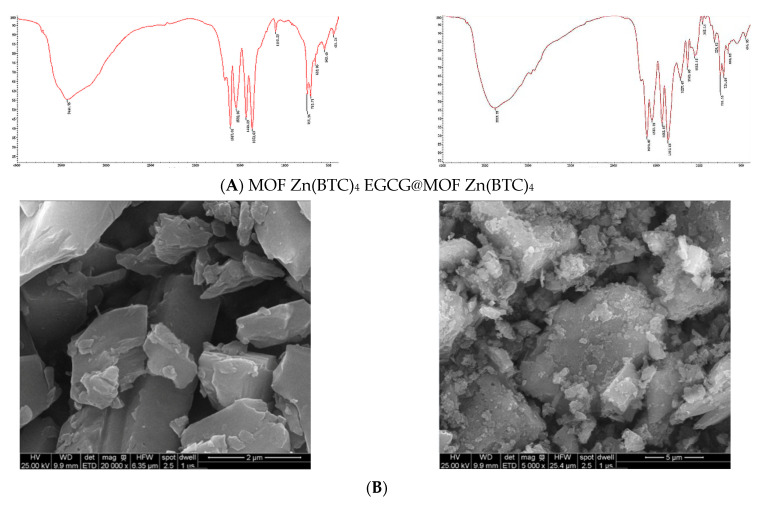
Comparison of MOF Zn(BTC)_4_ and EGCG@MOF Zn(BTC)_4._ (**A**) IR spectroscopy analysis of MOF Zn(BTC)_4_ and EGCG@MOF Zn(BTC)_4_; (**B**) Scanning electron microscopy (SEM) of MOF Zn(BTC)_4_ and EGCG@MOF Zn(BTC)_4_; (**C**) X-ray diffraction (XRD) analysis of MOF Zn(BTC)_4_ and EGCG_@_MOF Zn(BTC)_4_.

**Figure 6 molecules-27-05427-f006:**
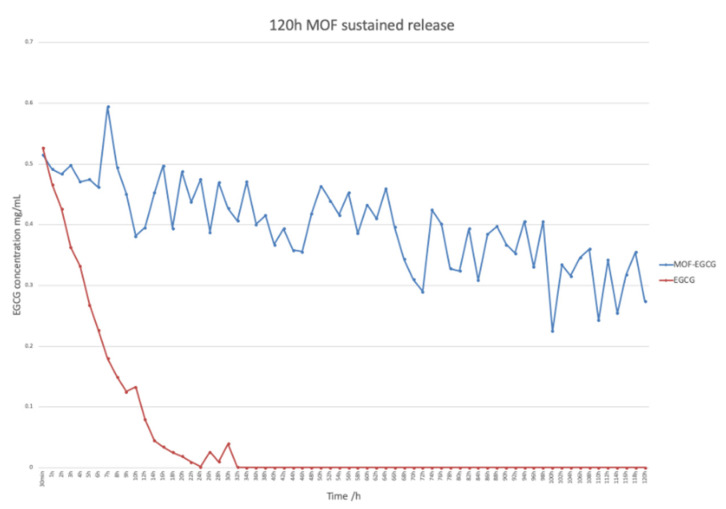
EGCG_@_MOF Zn(BTC)_4_ slows EGCG degradation.

**Figure 7 molecules-27-05427-f007:**
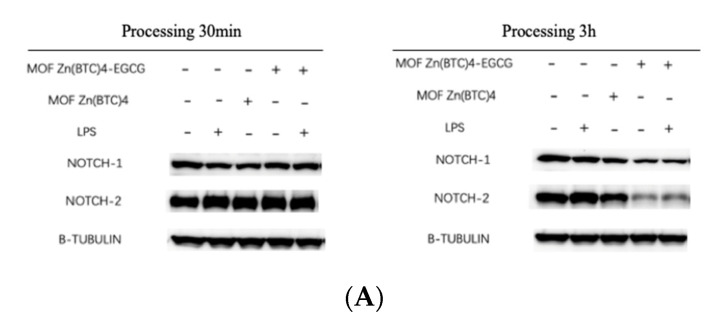
Effect of EGCG_@_MOF Zn(BTC)_4_ on Notch protein expression and inflammatory factors in macrophages. (**A**) Effect of EGCG@MOF Zn(BTC)_4_ on Notch protein expression at different time points; (**B**) Effect of EGCG@MOF Zn(BTC)_4_ on Notch protein expression at different time points; (**C**) Effect of EGCG@MOF Zn(BTC)_4_ on inflammatory factor levels.

**Figure 8 molecules-27-05427-f008:**
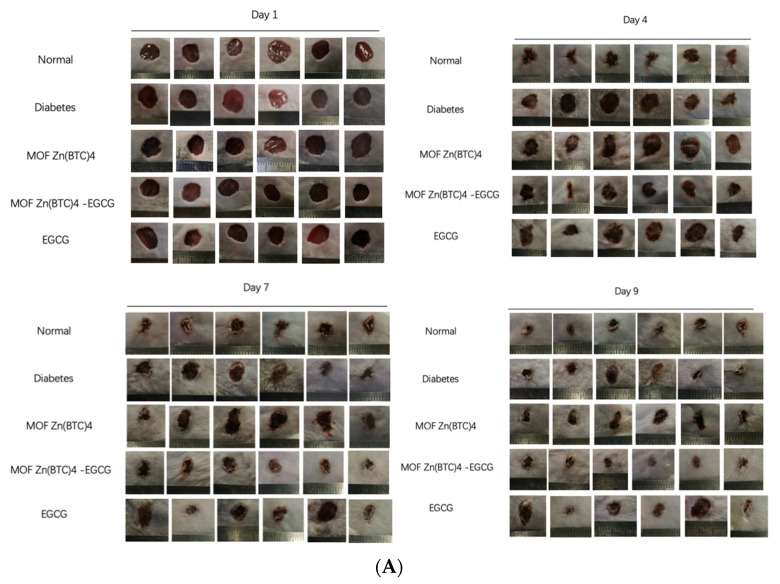
Wound healing and the broken line chart of the wound non-healing rate. (**A**) Wound conditions at different time points with treatment every 4 days; (**B**) Wound non-healing rate in the normal and model groups; (**C**) Wound non-healing rate, normal group vs. MOF Zn(BTC)_4_, *p* < 0.05; MOF Zn(BTC)_4_ vs. EGCG_@_MOF Zn(BTC)_4_, *p* < 0.05,B and C respectively indicate the arrogance between different groups.

**Figure 9 molecules-27-05427-f009:**
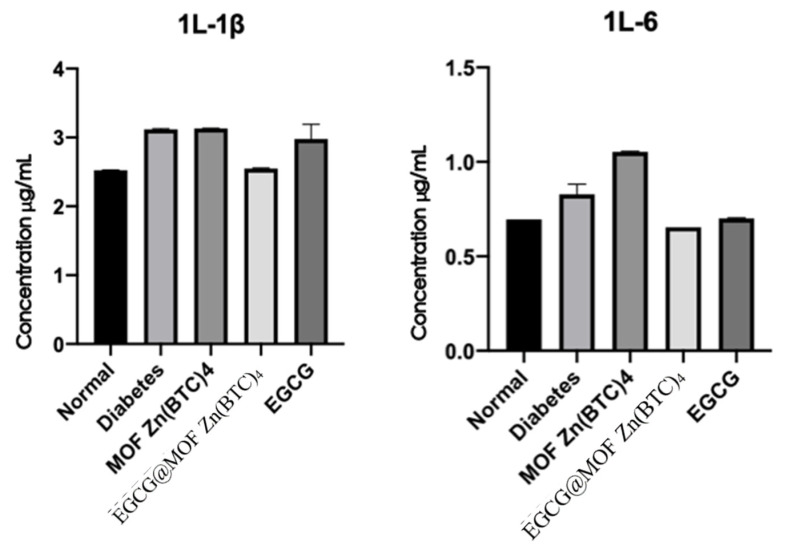
Expression of inflammatory factors IL-1β and IL-6 in wound tissue.

**Figure 10 molecules-27-05427-f010:**
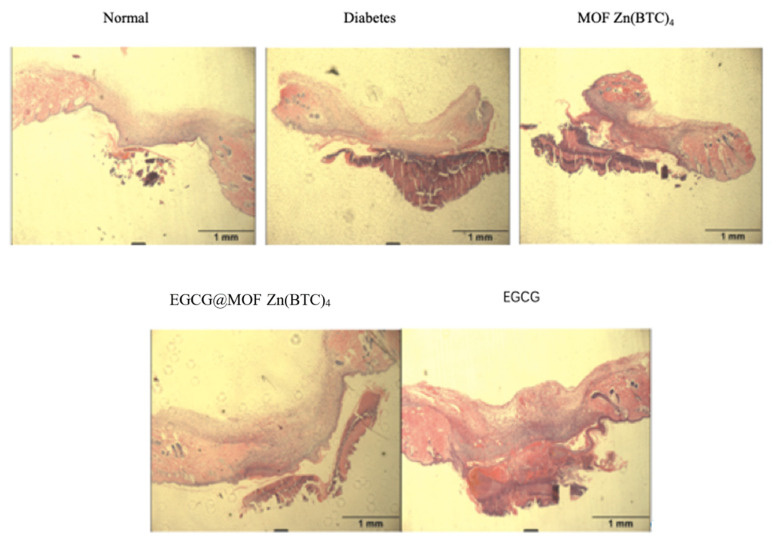
Tissue morphology in skin wounds.

**Figure 11 molecules-27-05427-f011:**
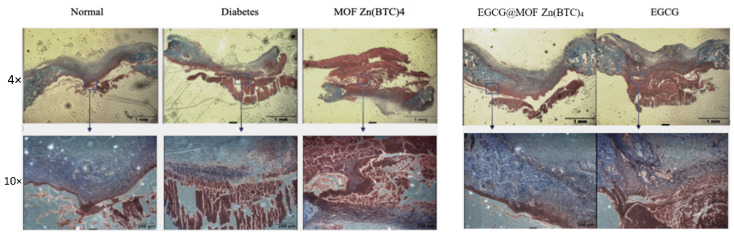
Collagen deposition.

**Table 1 molecules-27-05427-t001:** Crystal data of MOF Zn(BTC)_4_.

Empirical Formula	C_36_H_24_O_24_Zn	Z	4
Formula weight	809.67	μ/nm^−1^	1.403
Size/mm	0.17 × 0.15 × 0.10	Dc/(mg.cm^−3^)	1.449
θ rang for data collection/(°)	2.894 to 26.085	F(000)	808
Crystal system	Monoclinic	Reflections collected	9040
Space group	P2(1)/n	V/nm^3^	1795.7(2)
a/Å	9.5077(5)	Goodness of fit on F^2^	1.010
b/Å	16.3950(16)	R1,wR_2_(I > 2σ(I)	0.0456, 0.1080
c/Å	11.6119(9)	R1,wR_2_(all data)	0.0683, 0.1172
α/(°)	90.00	Δρmax(eÅ^−3^)	0.797
β/(°)	97.2200(10)	Δρmin(eÅ^−3^)	−0.544
γ/(°)	90.00		

**Table 2 molecules-27-05427-t002:** Selected bond lengths(Å) and angles(°) for MOF Zn(BTC)_4_.

Zn1 O5	1.939(3)	Zn1 O3	1.939(3)	Zn1 O2	1.967(3)
Zn1 O1	1.972(3)				
O5 Zn1 O3	125.79(12)		O5 Zn1 O2	94.56(12)	
O3 Zn1 O2	113.35(14)		O5 Zn1 O1	109.99(12)	
O3 Zn1 O1	99.47(12)		O2 Zn1 O1	114.57(12)	

## Data Availability

Not applicable.

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
