# Peer review of "Biological Effects of EGCG@MOF Zn(BTC)4 System Improves Wound Healing in Diabetes"

_molecules, 2022, doi:10.3390/molecules27175427_

Round 1

Reviewer 1 Report

The paper by Li et al. describes how MOF material (Zn(BTC)4) can be used to load natural biologically active compound epigallocatechin gallate (EGCG) and investigates the sustained release effect of Zn(BTC)4@EGCG and the biological effects on wound healing in a diabetic mouse model.

Although the study is well designed and data corroborate the conclusions, the paper should be improved before acceptance.

The authors should pay attention to the following issues:

·         Since EGCG is loaded into the Zn compound, the abbreviation EGCG@Zn(BTC)4 should be used instead of Zn(BTC)4@EGCG.

·         The authors did not perform a complete chemical characterization of the prepared “MOF”. There are at least 16 crystal structures of coordination polymers formed between Zn(II) and H3BTC. How can authors be sure that they obtained MOF instead of, for example, 1D coordination polymer? The authors are advised to overlap the powder XRD diffractogram of their sample with simulated powder XRD diffractograms of compounds formed between Zn(II) and H3BTC with known crystal structures (from CSD) to check if their sample has known structure. The sample needs proper chemical characterization if it is a novel compound.

·         Instrumentation data are missing (PXRD device, optical microscopy, SEM, and IR device).

·         Figure 2B is the same as figure 2A. Please give the proper figure 2B.

·         Part of the caption of fig. 6 is missing due to overlap with fig. 6C. Please prepare proper fig. 6.

·         Fig. 10 is misplaced.

·         A native English speaker should thoroughly check the text.

Author Response

Thank you very much for your help in our manuscript. As opinions given, we have already revised our manuscript point by point carefully and listed below.

Again, it is very nice of you to help us finish our work heartily. If there is anything else you need us to do, please tell us.

Best regard,

Dr. Li Song

The paper by Li et al. describes how MOF material (Zn(BTC)4) can be used to load natural biologically active compound epigallocatechin gallate (EGCG) and investigates the sustained release effect of Zn(BTC)4@EGCG and the biological effects on wound healing in a diabetic mouse model.

Although the study is well designed and data corroborate the conclusions, the paper should be improved before acceptance.

The authors should pay attention to the following issues:

  1. Since EGCG is loaded into the Zn compound, the abbreviation EGCG@Zn(BTC)4 should be used instead of Zn(BTC)4@EGCG.

Revision:Zn(BTC)4@EGCG has already been changed to EGCG@Zn(BTC)4.

  1. The authors did not perform a complete chemical characterization of the prepared “MOF”. There are at least 16 crystal structures of coordination polymers formed between Zn(II) and H3BTC. How can authors be sure that they obtained MOF instead of, for example, 1D coordination polymer? The authors are advised to overlap the powder XRD diffractogram of their sample with simulated powder XRD diffractograms of compounds formed between Zn(II) and H3BTC with known crystal structures (from CSD) to check if their sample has known structure. The sample needs proper chemical characterization if it is a novel compound.

Revision:Chemical characterization of MOF Zn(BTC)4 by crystal diffraction has been added in the results and discussion section of the article. 

  1. Instrumentation data are missing (PXRD device, optical microscopy, SEM, and IR device).

Revision:Instrumentation data are added in the part of General information

  1. Figure 2B is the same as figure 2A. Please give the proper figure 2B.

Revision:The figure 2B has been corrected in manuscript.

  1. Part of the caption of fig. 6 is missing due to overlap with fig. 6C. Please prepare proper fig. 6.

Revision:The position and size of Figure 6 have been adjusted correctly.

  1. Fig. 10 is misplaced.

Revision:Figure 10 has been examined and adjusted.

  1. A native English speaker should thoroughly check the text.

Revision:The manuscript was edited for correct English language usage, grammar, punctuation and spelling by qualified native English speaking editors at Charlesworth Author Services.

Thank you very much for your kindly and patiently review.

Reviewer 2 Report

This paper reported the sustained release effect of MOF Zn(BTC)4@EGCG and the biological effects on wound healing in a diabetic mouse model. I recommend publication in Molecules after minor revision based on the following comments.

1. In the introduction, it should be discussed detailed about the difference of MOFs materials with other materials.  

2. What is the effect of metals and ligands on release effect? critical discussions are required.

3. Figure 1 in this manuscript is not clear enough. The resolution needs to be improved.

4. More discussion should be included about the challenges and future advances in conclusions section.

5. The following papers should be taken into the appropriate sections in the revised version (especially in MOF applications in introduction).

a) Coordination Chemistry Reviews 469 (2022) 214664, b) Inorg. Chem. 61 (2022) 9514–9522, c) Journal of Physics: Energy 3 (2021) 032010, d) Coordination Chemistry Reviews 461 (2022) 214505, e) New J. Chem. 46 (2022) 9440-9450, f) Inorg. Chem. 61 (2022) 3396–3405.

6. The writing should be improved.

7. What happens if you just mix the MOF Zn(BTC)4 + EGCG for the biological effects on wound healing in a diabetic mouse model?

Author Response

Thank you very much for your help in our manuscript. As opinions given, we have already revised our manuscript point by point carefully and listed below.

Again, it is very nice of you to help us finish our work heartily. If there is anything else you need us to do, please tell us.

Best regard,

Dr. Li Song

This paper reported the sustained release effect of MOF Zn(BTC)4@EGCG and the biological effects on wound healing in a diabetic mouse model. I recommend publication in Molecules after minor revision based on the following comments.

  1. In the introduction, it should be discussed detailed about the difference of MOFs materials with other materials.  

Revision:The discussion of the differences of MOFs materials with other materials has been added in the introduction part.

  1. What is the effect of metals and ligands on release effect?critical discussions are required.

Revision:The discussion of metals and ligands on release effect has been accomplished in the introduction part.

  1. Figure 1 in this manuscript is not clear enough. The resolution needs to be improved.

Revision:The definition of Figure 1 has been enhanced.

  1. More discussion should be included about the challenges and future advances in conclusions section.

Revision:The discussion about the challenges and future advances has been accomplished in conclusions section.

  1. The following papers should be taken into the appropriate sections in the revised version (especially in MOF applications in introduction).
  2. a) Coordination Chemistry Reviews 469 (2022) 214664, b) Inorg. Chem. 61 (2022) 9514–9522, c) Journal of Physics: Energy 3 (2021) 032010, d) Coordination Chemistry Reviews 461 (2022) 214505, e) New J. Chem. 46 (2022) 9440-9450, f) Inorg. Chem. 61 (2022) 3396–3405.

Revision:Corresponding references have been added to the manuscript.

  1. The writing should be improved.

Revision:The manuscript was edited for correct English language usage, grammar, punctuation and spelling by qualified native English speaking editors at Charlesworth Author Services.

  1. What happens if you just mix the MOF Zn(BTC)4 + EGCGfor the biological effects on wound healing in a diabetic mouse model?

Reply:MOF is a new type of porous organic-inorganic hybrid functional material with large specific surface area and porosity. The pore size can store and release drugs with a molecular weight smaller than the pore size of the skeleton. Its functional groups can also interact with the drug functional groups to achieve the purpose of drug loading.

Thank you very much for your kindly and patiently review.

Reviewer 3 Report

The authors presented Biological effects of MOF Zn(BTC)4@EGCG system improves

wound healing in diabetes. The work is interested and can be accepted but the following comments must be considered before production

My comments

1.    Aim of the work should be stated clearly in introduction

2.    For Metal-organic frameworks (MOFs) are a new type of porous material that combines inorganic and organic component, the following citation should be added: Inorganic Chemistry Communications 138, 2022, 109251

3.     Resolution of figures  must be enhanced

4.     The current study should be compared with other related studies in literature

5- The authors must revise language of the manuscript before publication and the whole article must be adjusted based on journal style.

Author Response

Thank you very much for your help in our manuscript. As opinions given, we have already revised our manuscript point by point carefully and listed below.

Again, it is very nice of you to help us finish our work heartily. If there is anything else you need us to do, please tell us.

Best regard,

Dr. Li Song

The authors presented Biological effects of MOF Zn(BTC)4@EGCG system improves

wound healing in diabetes. The work is interested and can be accepted but the following comments must be considered before production

My comments

  1. Aim of the work should be stated clearly in introduction

Revision:Aim of the work has been clearly given in introduction.

  1. For Metal-organic frameworks (MOFs) are a new type of porous material that combines inorganic and organic component, the following citation should be added: Inorganic Chemistry Communications 138, 2022, 109251

 Revision:Corresponding references have been added to the manuscript.

  1. Resolution of figures  must be enhanced

 Revision:All pictures have been enlarged and clarified.

  1. The current study should be compared with other related studies in literature.

Revision:The current related study has been compared in introduction part.

5. The authors must revise language of the manuscript before publication and the whole article must be adjusted based on journal style.

Revision:The manuscript was edited for correct English language usage, grammar, punctuation and spelling by qualified native English speaking editors at Charlesworth Author Services.

Thank you very much for your kindly and patiently review.